# Species Differences and Tissue Distribution of Heavy Metal Residues in Wild Birds

**DOI:** 10.3390/ani14020308

**Published:** 2024-01-18

**Authors:** Patamawadee Khwankitrittikul, Amnart Poapolathep, Saranya Poapolathep, Chayanid Prasanwong, Sittinee Kulprasertsri, Kraisiri Khidkhan

**Affiliations:** 1Department of Pharmacology, Faculty of Veterinary Medicine, Kasetsart University, Bangkok 10900, Thailand; patamawadee.kh@ku.th (P.K.); amnart.p@ku.th (A.P.); saranya.po@ku.th (S.P.); 2Department of National Parks, Wildlife and Plant Conservation, Bangphra Waterbird Breeding Center, Bangphra, Sriracha, Chonburi 20110, Thailand; vet.noknam@gmail.com; 3Department of Farm Resources and Production Medicine, Faculty of Veterinary Medicine, Kasetsart University, Kamphaeng Saen Campus, Nakhon Pathom 73140, Thailand; jubjubsk@gmail.com

**Keywords:** bioindicators, biomonitoring study, interspecies differences, tissue accumulation, toxic metals

## Abstract

**Simple Summary:**

This study aimed to evaluate heavy metal deposits in the feathers, livers, and kidneys of 37 wild bird species in Thailand. Interspecies differences in metal residues were noticed in the livers and kidneys but not in the feathers. The species-specific circumstances of the granivorous birds, especially the feeding behavior, and/or the ingestion of metal-contaminated seeds may result in a higher accumulation of Cd in their livers and kidneys compared to those in water birds. The Pb concentration in the livers (>15 ppm) and feathers (>4 ppm) exceeded the thresholds, causing potential lead poisoning and affecting the reproductive success in all the groups of birds. The Cd level in the kidneys was above 2–8 ppm, indicating increased environmental exposure to Cd in these wild birds. The correlations showed the suitability of the liver or kidney as an alternative sample for each other in studying the accumulation of Cd, Pb, Ni, Zn, and Fe, while Pb in the liver could be estimated using feathers.

**Abstract:**

Birds are useful as bioindicators of metal pollution, but the variety of species and tissue distribution may influence the study of heavy metal burdens in birds. The objective of this study was to determine the levels of heavy metals in wild birds’ carcasses to acquire information on species differences and the tissue distribution of metals in wild birds in Thailand. Species differences in metal buildup were observed in the livers and kidneys, but not in the feathers. A significantly higher accumulation of Cd was found in the livers and kidneys of the granivorous birds compared to those in the water birds. In all the groups of birds, the Pb level in the livers (>15 ppm) and feathers (>4 ppm) exceeded the threshold limits, causing potential lead poisoning and disturbing the reproductive success. The Cd accumulation in the kidneys was above 2–8 ppm, indicating increased environmental exposure to Cd in these birds. The Cd, Pb, Ni, Zn, and Fe concentrations in the livers could be estimated using the kidneys, while the Pb level in the liver may be predicted using feathers. Furthermore, water birds’ feathers may be potentially appropriate bioindicators for long-term exposure. Research on the origin of metal contamination is needed to reduce the threat of heavy metals to the health of both birds and other wildlife species.

## 1. Introduction

Heavy metals are persistent toxic substances that are widespread in the environment and can accumulate and biomagnify throughout the food web [1]. Certain metals, including zinc (Zn), iron (Fe), copper (Cu), and manganese (Mn), are necessary for organisms but may be harmful in large concentrations, while nonessential metals, such as lead (Pb), cadmium (Cd), nickel (Ni), and mercury (Hg), have negative effects on animals even at low concentrations [2]. Heavy metal pollution in aquatic and terrestrial ecosystems is a global environmental concern because it threatens structural function, behavior, population, and diversity at many trophic levels [3]. 

Birds are more sensitive to changes in the ecosystem compared to other species because of their behavioral and physiological characteristics [4,5]. They are widely distributed and can be exposed to heavy metals from the environment by several routes, such as ingesting contaminated foods and water, inhaling polluted air, and dermal contact, resulting in the accumulation of large amounts of elements in their body tissues such as the lungs, liver, kidneys, skin, and feathers [4,6,7,8]. Therefore, the accumulation of heavy metals in the bird’s organs can reflect the presence of pollutants in the environment [9,10]. In addition, heavy-metal-disturbed avian physiological processes and behavioral traits are easily noticeable. For example, Pb poisoning can impair immune system function and damage the nervous system, and Cd can reduce growth performance and reproduction [6,11,12,13]. For these reasons, using birds as bioindicators of environmental pollution is becoming more widespread. 

Although birds live in the same geographical area, their genetic and physiological characteristics, residency patterns, and dietary and feeding behaviors are different [14]. Previous studies suggested that feeding strategy, feeding habit, and type of food items consumed significantly affect metal accumulation in avian tissues [15,16]. Ittiporn et al. (2012) [17] reported that herbivorous birds, especially Scarlet-backed Flowerpecker (*Dicaeum cruentatum*), showed the highest Pb level in feathers followed by carnivores and omnivores. In addition, insectivorous birds (Barn Swallow; *Hirundo rustica*) and insectivore–granivores (House Sparrow; *Passer domestics*) revealed a great tendency to accumulate heavy metals (Pb, Cd, Cu, Zn, and Mn) in their tissues (muscle, liver, kidney, and feathers) compared to water bird species (Great Egret; *Ardea alba* and Striated Heron; *Butorides striata*) [18]. On the other hand, a biomonitoring study in Pakistan using bird feathers found that the accumulation of metals (Cd, Pb, Ni, Cu, Mn, Zn, and Fe) was greatest in carnivorous birds followed by omnivorous and insectivorous species, and granivores showing minimal levels [16]. 

Several tissues are used to evaluate element concentrations in birds, but the most frequently analyzed tissues for toxic metals are livers, kidneys, and feathers [19]. In general, the accumulation of heavy metals reveals tissue-specific differences because of varied environmental circumstances and organ functions [20,21,22]. El-Shabrawy et al. (2022) [18] found that the kidney is the tissue most burdened by heavy metals compared to liver and feather of the insectivore (Barn Swallow; *Hirundo rustica*) and water birds (Great Egret; *Ardea alba* and Striated Heron; *Butorides striata*). Furthermore, Ashraf and Ali (2022) [23] reported that the highest concentrations of Cd, Pb, Ni, Mn, Cu, and Zn were found in the intestine followed by the gizzard and feather of water bird species including Eurasian Coot (*Fulica atra*), Gadwall (*Anas strepera*), and Common Teal (*Anas crecca*). In Cattle Egrets (*Bubulcus ibis*), the highest levels of Cd, Pb, Ni, Cu, and Zn were reported in the liver followed by muscle and feathers [24]. In contrast to the recent study from Lahel et al. (2023) [25], the highest level of Pb was detected in the feathers of Great Cormorant (*Phalacrocorax carbo*) followed by the kidney, liver, and bone, whereas the highest burden of Cd was found in the kidney followed by the bone, liver, and feather. However, a biomonitoring study in Pakistan found that there was no evidence of organ preference (liver, lung, kidney, feather, and brain) for the accumulation of metals (Cd, Cu, Mn, Cr, Co, and Zn) in examined granivorous, omnivorous, and carnivorous birds [26].

The use of feathers for assessing the contamination of heavy metals in birds is becoming popular since it is suggested as a non-destructive way that causes little damage to the population of, and individual, birds and is useful for long-term study [27,28]. However, the use of feathers in place of visceral tissues to monitor heavy metals is still unclear. In raptor species, Castro et al. (2011) [29] found a significant correlation for the Cd level between the feathers and kidneys of Northern Goshawk (*Accipiter gentilis*) and Tawny Owl (*Strix aluco*) and a significant correlation for Cd and Pb concentrations between the feathers and livers of the Common Buzzard (*Buteo buteo*). In insectivore–granivorous birds, Ding et al. (2023) [29] suggested that Pb and Cd in the internal organs (liver, kidney, heart, and lung), skeletons, and muscles of Eurasian Tree Sparrow (*Passer montanus*) can be estimated using feathers. Similar to the House Crow (*Corvus splendens*), an omnivorous bird, this revealed significant positive correlations for the Pb level between feathers and internal organs (heart, lung, brain, liver, and kidney) [9]. On the other hand, in water birds (White-breasted Waterhen; *Amaurornis phoenicurus* and Common Moorhen; *Gallinula chloropus*), no significant correlation for the Cd, Pb, Cu, and Ni concentrations among the visceral tissues (heart, liver, and kidney) and feathers was observed by Mukhtar et al. (2020) [30].

Based on the results of earlier studies [9,16,17,18,24,29], we hypothesized that there are differences in the heavy metal accumulation among different types of bird species. Herbivorous, carnivorous, or insectivorous birds may accumulate heavy metals more than other groups of birds. In addition, we predicted that the kidney may be the target organ of heavy metals compared to other tissues. The possibility of predicting metal levels in internal organs (i.e., liver and kidney) from the concentrations in feathers was also estimated in this study. Therefore, the objectives of this study were to determine the levels of heavy metals (Cd, Cu, Fe, Ni, Pb, and Zn) in the feathers, livers, and kidneys of wild birds, and to evaluate the correlation of these tissue samples for each metal. These findings will offer data on species differences and the tissue distribution of heavy metals that may support the use of these birds as appropriate biosentinels of dangerous metal contamination in environments and the potential for using feather samples to monitor heavy metal body burdens in wild birds.

## 2. Materials and Methods

### 2.1. Study Site and Sample Collection 

This experiment was authorized under permits from the Department of National Parks, Wildlife, and Plant Conservation. All procedures related to animal use were performed according to the Guidelines for Animal Experiments and approved by the Animal Ethics Research Committee of the Faculty of Veterinary Medicine, Kasetsart University (approval ID: ACKU65-VET-033). Sixty carcasses of wild birds were collected between September 2022 and July 2023 in the Bang Phra Waterbird Breeding Center, an animal hospital and bird research facility situated in the Bang Phra Reservoir Nonhunting Area, Chonburi province, Thailand (Figure 1). These avian species were killed by accidents and injuries. However, veterinarians did not report any pathological changes or other lesions in these collected birds to diagnose other diseases. They were protected wild birds in the Bang Phra Reservoir Nonhunting Area and were classified by feeding behavior and habitat [31,32,33,34,35,36,37,38,39,40] into six categories: water birds (*n* = 24), predatory birds (*n* = 12), omnivores (*n* = 8), insectivore–frugivores (*n* = 5), insectivores (*n* = 5) and granivores (*n* = 6). Based on birds’ diet information [36,40], some bird species commonly eat both insects and small fruits, while the daily nourishment of some birds is only insects. Therefore, groups of insectivore–frugivores and insectivores were completely separated in this study [36,39]. Population status of the studied birds was classified by IUCN [41]. The details of 37 species (*n* = 60) are shown in Table 1. Veterinarians dissected all the bird carcasses. The livers, kidneys, and feathers (pooled primary and secondary feathers from the wings, legs, breast, neck, and tail) were collected from each bird and kept at −80 °C.

### 2.2. Sample Preparation and Heavy Metal Determination

Sample preparation for heavy metal analysis in this study was modified from previous studies [9,42]. Feather samples were washed with deionized water and cut into small pieces. All types of samples, including livers, kidneys, and feathers were dried in a hot air oven at 60 °C for 24 h. A total of 0.5 g of each sample was weighed and digested in a solution (1:1 *v*/*v*) containing 65% nitric acid (Merck, Darmstadt, Germany) and 30% hydrogen peroxide (Chem-Supply, Gillman, SA, Australia). Then, the samples were diluted by adding deionized water up to 25 mL and analyzed for the concentrations of heavy metals (Cd, Pb, Ni, Cu, Zn, and Fe) using a flame atomic absorption spectrophotometer (FAAS; 240 AA Agilent Technologies, Santa Clara, CA, USA).

### 2.3. Assessment and Quality Control

A procedural blank was created along with the samples to assess any potential metal contamination throughout sample preparation and analytical processes. The method validation was assessed by spiking 0.1, 0.5, and 1 ppm of the six elements into the three sample matrices of day-old chicks. The spike recoveries of the six metals varied from 82% to 113% in the feather, from 84% to 119% in the liver, and from 84% to 121% in the kidney. The analytical sensitivity of the FAAS was assessed by determining the values of the limit of detection (LOD) and the limit of quantification (LOQ). The LODs were 0.0004 ppm for Cd and Cu, 0.001 ppm for Pb and Zn, 0.003 ppm for Ni, and 0.01 ppm for Fe. The LOQs were 0.001 ppm for Cd and Cu, 0.004 ppm for Pb, 0.005 ppm for Zn, 0.01 ppm for Ni, and 0.04 ppm for Fe. All samples were run in triplicate for each analytical run.

### 2.4. Statistical Analysis

All heavy metal concentrations (ppm) in the feathers, livers, and kidneys were calculated on a dry weight basis. Feather thresholds for Cd, Pb, and Zn toxicity are 2 ppm, 4 ppm, and 1200 ppm, respectively [7,43]. The threshold levels in the livers of birds are 40 ppm for Cd poisoning and >15 ppm for severe Pb poisoning [44]. In kidneys, the avian threshold of increased environmental Cd exposure is 2–8 ppm and the limited level of Pb associated with impaired survival is >15 ppm [45]. Descriptive statistics of the data were presented as the mean ± standard deviation (SD), median, minimum, and maximum values. The JMP version 17 (SAS Institute, Cary, NC, USA) was used for the statistical analyses and graphical data creation. The levels of heavy metal among tissues and groups of birds were tested for normality using the Shapiro–Wilk test and determined for homogeneity of variance using Levene’s test. The Kruskal–Wallis test and Dunn test (post hoc multiple comparisons) were used to examine variations in heavy metal concentrations across groups of birds and between bird tissues. Correlation analysis (Spearman’s correlation) was performed to understand the relationship between feathers, livers, and kidneys for different heavy metals. A *p* value of <0.05 was deemed to signify statistical significance in all analyses.

## 3. Results

### 3.1. Tissue Distribution of Heavy Metals

The mean (±SD) and median of the heavy metal concentrations of all the bird species indicate that the trend of the six metal accumulations in the tissues was as follows: feathers: Zn > Fe > Pb > Cu > Ni > Cd; livers: Fe > Zn > Pb > Cu > Ni > Cd; kidneys: Fe > Zn > Pb > Ni > Cu > Cd (Table 2). The maximum concentrations of all the elements were found in the kidneys (Cd, Pb, Ni, and Zn) and livers (Cu and Fe), whereas the minimum levels of those metals were observed in the feathers. The mean concentrations of Cd, Pb, Ni, and Zn in the kidneys were significantly higher than those found in the liver (Cd; *p* < 0.0001, Pb; *p* = 0.0003, Ni; *p* < 0.0001, and Zn; *p* = 0.0022) and feather samples (Cd; *p* < 0.0001, Pb; *p* < 0.0001, Ni; *p* < 0.0001, and Zn; *p* = 0.0036) (Figure 2). In the livers, the mean concentrations of all the metals (Cd, Pb, Ni, and Fe), except Cu (*p* = 0.4998) and Zn (*p* = 1.0000), were significantly higher than those found in the feathers (Cd; *p* < 0.0001, Pb; *p* < 0.0001, Ni; *p* < 0.0001, and Fe; *p* < 0.0001). The mean level of Fe in the feathers was significantly lower than those detected in the livers (*p* < 0.0001) and kidneys (*p* < 0.0001), while there was no significant difference in the Fe concentration between the livers and kidneys (*p* = 0.0772). Nevertheless, substantial variations in the Cu content across all the tissues were not observed (*p* = 0.1857–1.0000).

Although specific patterns of the heavy metal accumulations were observed in the tissues, the proportions of heavy metal concentrations in the same tissues were almost similar among the six groups of birds (Figure 3). Comparing the relative concentrations of the elements in each group of birds, Fe (72.6–90.2%) was the highest-level metal found in the livers, while the largest accumulations of Zn (35.2–60.5%) and Cu (2.2–7.3%) were found in the feathers of six groups of birds. Conversely, the highest proportional concentrations of Cd (0.3–2.1%), Pb (7.5–27.4%), and Ni (1.1–5.2%) were observed in the kidney tissues of all the groups of birds.

### 3.2. The Differences of Metal Residues Were Related to the Groups of Birds

In feather samples, the highest concentrations of Cd (1.6 ppm), Pb (32.1 ppm), and Zn (488.4 ppm) were found in the Red-wattled Bulbul, while the maximum levels of Ni (4.5 ppm), Cu (279.9 ppm), and Fe (1053.2 ppm) were found in the Asian Openbill, Watercock, and Oriental Scops Owl, respectively. However, the average levels of all the elements in the feather tissues were not significantly different among the six groups of birds (Figure 4A). In the livers, the concentration of Cd was much higher in the granivores compared to those in the water birds (*p* = 0.0243), but there was no discernible difference in the concentrations of the other heavy metals (Pb, Ni, Cu, Zn, and Fe) among the six groups of birds (Figure 4B). The maximum concentrations of Pb (254.1 ppm) and Zn (703.7 ppm) were detected in the Zebra Dove, whereas the highest accumulations of Cd (8.4 ppm), Ni (17.3 ppm), Cu (1021.8 ppm), and Fe (5088.4 ppm) were observed in the Thick-billed Green Pigeon, Large-tailed Nightjar, Greater Painted-snipe, and Watercock, respectively. In the kidney tissues, a significantly higher concentration of Cd was found in the granivores compared to those in the water birds (*p* = 0.0442) (Figure 4C). The comparison of the heavy metal concentrations in the kidney of the insectivore–frugivores was excluded from this result because it could be collected only from one sample from the Lineated Barbet. However, there was no significant difference in the concentrations of the other heavy metals (Pb, Ni, Cu, Zn, and Fe) in the kidneys among five group of birds. The maximum concentrations of Cd (64.1 ppm) and Cu (71.5 ppm) were observed in the Asian Koel and the maximum levels of Pb (2303.1 ppm) and Ni (311.2 ppm) were detected in the Common Myna, whereas the highest accumulations of Zn (755.9 ppm) and Fe (2020.8 ppm) were found in Gurney’s Pitta and the Watercock, respectively. 

### 3.3. The Relationship of Accumulated Tissues for Heavy Metals

Significant correlations (*p* < 0.05) were found between the liver tissues and kidney tissues for the accumulations of Cd (*ρ* = 0.7988, *p* < 0.0001), Pb (*ρ* = 0.6253, *p* < 0.0001), Ni (*ρ* = 0.5468, *p* < 0.0001), Zn (*ρ* = 0.5241, *p* = 0.0002), and Fe (*ρ* = 0.3323, *p* = 0.0257) (Table 3). Furthermore, the concentration of Pb in the feathers was positively correlated with that in the livers (*ρ* = 0.4874, *p* = 0.0003), but a negative correlation was found between the feathers and kidney tissues for the Cu concentrations (*ρ* = −0.3486, *p* = 0.0189).

## 4. Discussion

The heavy metal burdens in all the studied birds showed that there are differences in the trend of element buildups among bird species and tissues. The maximum levels of most elements were found in the kidneys and livers, but the lowest accumulations of those metals were observed in the feathers. The granivorous birds revealed a higher Cd accumulation in the livers and kidneys in comparison with the water birds. Furthermore, significant correlations were found between the liver tissues and kidney tissues for the concentrations of most metals. Comparing the variations in the metal residues among the tissues, the highest concentrations of most elements (Cd, Pb, Ni, and Zn), and the maximum fractions of Cd, Pb, and Ni found in the kidneys of all the bird species suggested that the kidney could be a target accumulation site for those metals, except for Fe and Cu. Although the kidney samples of the Common Myna and Asian Koel showed very high levels of Cd and Pb, gross lesions and other infectious diseases were not diagnosed in those samples in this study. Since the liver is the primary organ of red blood cell removal and Fe recycling [46], the highest concentration of Fe was observed in the livers of the birds in our study. Although the functions of the kidney and liver are closely intertwined and both organs are sites of metal detoxification in birds [9], the results indicated that the kidney is the organ accumulating most heavy metals compared to liver. However, the significant correlations of Cd, Pb, Ni, Zn, and Fe between the liver and kidney tissues in the present study suggested the suitability of the liver or kidney as a substitute sample for each other in studying metal buildup in birds.

In all the bird species, the similar levels of some heavy metals (Cu and Zn) found in the feathers and livers and the findings of the positive correlation of Pb concentrations between the feathers and livers suggested that Pb accumulation in liver tissues could be estimated using feather samples. However, the significant correlation of metal accumulations between the feathers and kidneys was not found in the current investigation. Gruz et al. (2019) [8] and Varagiya et al. (2021) [7] suggested that feather analysis is not an alternative to internal tissue analysis. It ought to be viewed as a precursor to the detrimental consequences of heavy metal exposure in birds. 

Bird feathers are the sites of heavy metal accumulation and excretion and are considered significant bioindicators for long-term exposure to environmental pollution [8,9]. Earlier research has reported that Zn and Cu are essential metals accumulated in feathers of most predatory birds, including the Common Buzzard, Barn Owl, Eurasian Sparrowhawk, Osprey, and Common Kestrel [47,48]. Similar to our results, in addition to predatory birds, the highest proportions of Zn and Cu were detected in the feathers of the other groups of birds including the water birds, omnivores, insectivore–frugivores, insectivores, and granivores. These metals might have a preference for the pigments that give birds their coloring, for example, the darker black and iridescent blue feathers contained considerable amounts of Fe and Cu, while Zn was noticed in the iridescent feathers [48,49]. The feather Cd levels of the six groups of birds in the present study were below the threshold level (2 ppm), but the Pb concentrations were critically above the threshold (4 ppm) that affects reproductive success in birds [7,43].

Interestingly, there is no discernible difference in all the metal accumulations in the feathers among the six groups of birds, but the maximum concentrations of the heavy metals (i.e., Cd, Pb, Ni, Cu, and Zn) in the feather samples were observed in the water bird species, including the Red-wattled Lapwing, Asian Openbill, and Watercock. Unlike the other groups of birds, we observed that only these water birds had higher levels of Cd, Pb, Cu, and Zn in their feathers than in their livers and kidneys, which were similar to an earlier report in other water birds: the Black-crowned Night Heron and Indian Pond Heron [50]. According to the data, these water bird species may specially store and excrete those metals through their feathers, and their feathers may be appropriate bioindicator samples for long-term exposure to the environment. However, further biomonitoring studies and the increased sample sizes are required to better understand the differences in the accumulation patterns of metals in feathers, which could be an initial warning of the toxic effects on water birds and other bird species.

Species-specific genetic characteristics and behavioral traits play an important role in the possibility of contaminant exposure [14]. The present study indicated that there are some specific conditions in the group of granivores, such as feeding habits and/or genetic and physiological traits, that cause a higher accumulation of heavy metals in their livers and kidneys compared to those in water birds. Avian species are exposed to heavy metals through direct ingestion, inhalation, and dermal contact absorption, but food is considered the most significant route in wildlife [4,12,14]. In our study, only two species, the Zebra Dove and Thick-billed Green Pigeon, were classified into the group of granivores [36,39]. Although granivorous birds feed on various seeds, droplets, and fruits of plants, they are highly selective in their seed choice, which is influenced by bill structures [51,52,53,54]. The preferred seeds of these bird species may be highly contaminated with heavy metals, posing health risks to the granivores ingesting those seeds as the principal component of their diet. In addition, Zebra Doves forage on the ground by walking and pecking at grass seeds and grains that they find in their surroundings [55]. As a result of this feeding habit, this bird species may be easily exposed to heavy metals in soils and could be a good biosentinel species for heavy metal contamination on the ground compared to other groups of birds. 

The interspecies differences in the metal deposits in the livers and kidneys but not feathers in our study suggested that the selection of tissue samples is crucial for determining heavy metal toxicity in birds. The greater accumulation of Cd in the kidney and liver samples of the granivorous birds compared to those in the water birds may result in a higher potential toxicity risk of heavy metals to the granivorous bird species (e.g., Thick-billed Green Pigeon and Zebra Dove). In addition to the granivores, the water birds, predatory birds, omnivores, insectivore–frugivores, and insectivores had a mean Pb level in their livers above the cutoff point (>15 ppm) for the identification of severe clinical lead poisoning in birds [44]. Lead poisoning in birds not only causes severe clinical signs (i.e., impaired mobility, lowered sensorial ability, anemia, and ataxia) but may cause birds to be more susceptible to other diseases and predation [56]. For Cd, the findings exceeded the reference range [45] in the kidneys (2–8 ppm), indicating the increased environmental exposure to Cd in most of the protected wild birds in this study. 

This study includes some limitations. For example, the number of species differed considerably among the six groups of birds, and the sample size of each group was small. The increased sample sizes for all the groups of birds are required to better understand the differences in the deposition patterns of the metals in these bird species. Previous studies have revealed gender and age differences in the accumulation of heavy metals in birds [57,58]; nonetheless, we did not consider sex-depended differences in metal accumulation, and most of the tissue samples were collected from adult birds. In addition to gathered bird organs, the analyses of other environmental samples such as soil, water, and bird feed are also critical to identify the sources and pathways of heavy metal contamination in these wild birds. Additionally, more studies on the toxicity and susceptibility of heavy metals in these birds are also needed.

## 5. Conclusions

Interspecies differences in the metal residues were noticed in the livers and kidneys but not in the feathers. The species-specific conditions and feeding habits of the granivorous birds and/or the ingestion of heavy metal-contaminated seeds may result in a higher accumulation of most metals in their livers and kidneys compared to those in the water birds. The concentration of Pb in the livers (>15 ppm) and feathers (>4 ppm) exceeded the threshold levels, causing potential lead poisoning and reproductive failure risks to affect the six groups of the birds. The Cd accumulation in the kidneys was above 2–8 ppm, indicating increased environmental exposure to Cd in these wild birds. The correlations indicated the suitability of the liver or kidney as an alternative sample for each other in studying the accumulation of Cd, Pb, Ni, Zn, and Fe, while only the Pb in the livers could be estimated using the feathers. In addition, some species of water birds (i.e., Red-wattled Lapwing, Asian Openbill, and Watercock) may considerably accumulate and excrete toxic metals through their feathers, and their feathers may be potentially suitable bioindicators for long-term exposure to the environment. Overall, the study revealed an alarmingly high concentration of hazardous metals in the wild birds, which serves as a warning about metal pollution in the environment.

## Figures and Tables

**Figure 1 animals-14-00308-f001:**
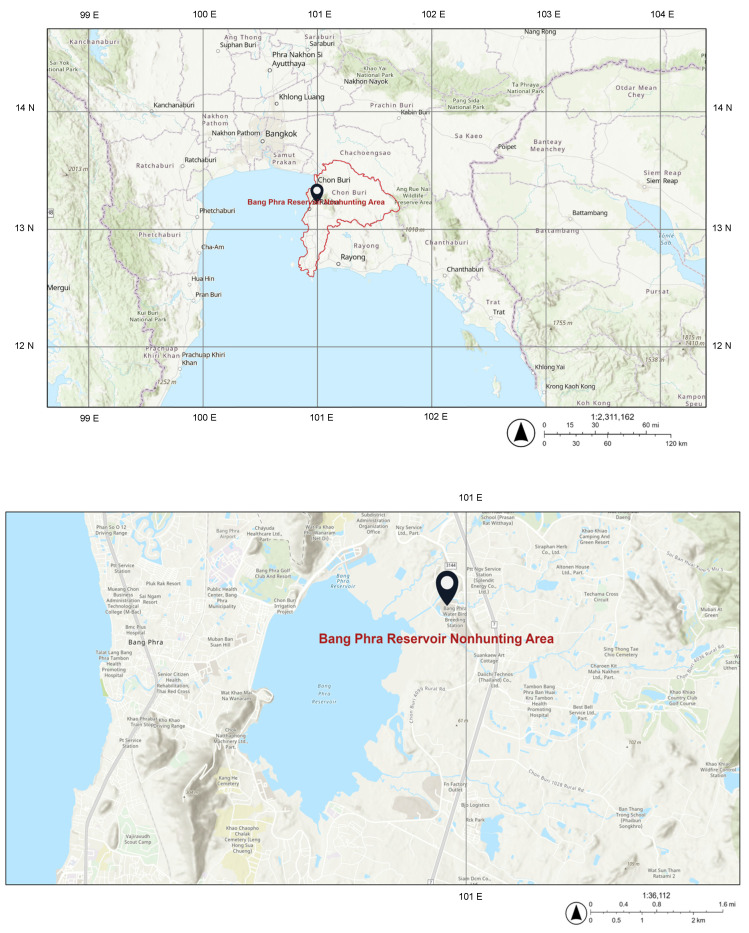
The location of sample collection (Bang Phra Reservoir Nonhunting Area (13.1585° N, 101.0176° E), Chonburi province (13.2017° N, 101.2524° E), Thailand).

**Figure 2 animals-14-00308-f002:**
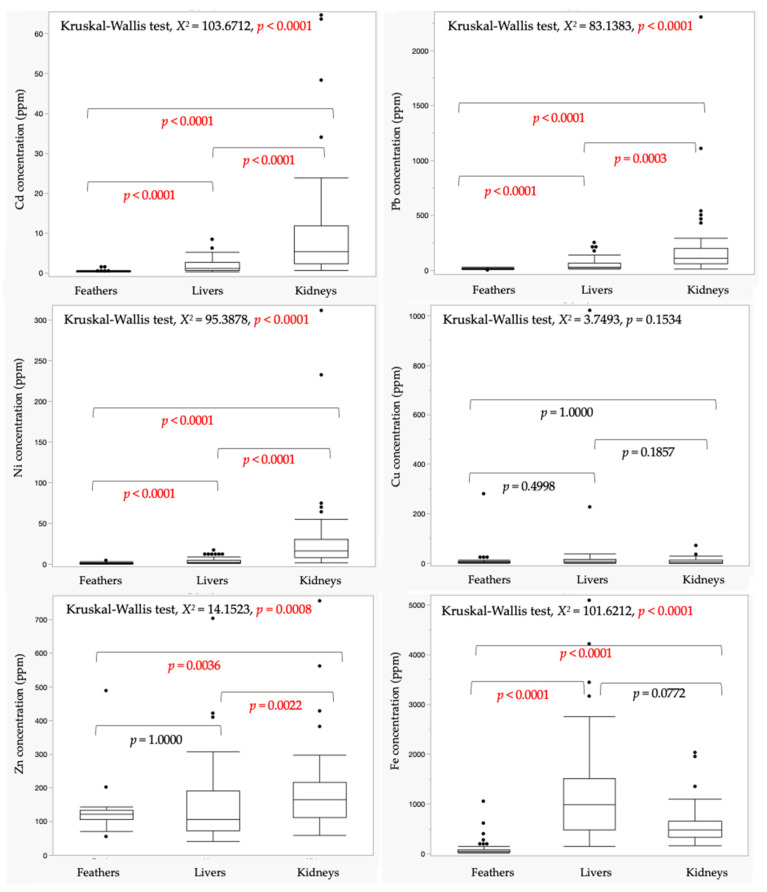
Boxplot comparison of heavy metal concentrations in feathers, livers, and kidneys of all birds. The *p* < 0.05 was considered statistically significant.

**Figure 3 animals-14-00308-f003:**
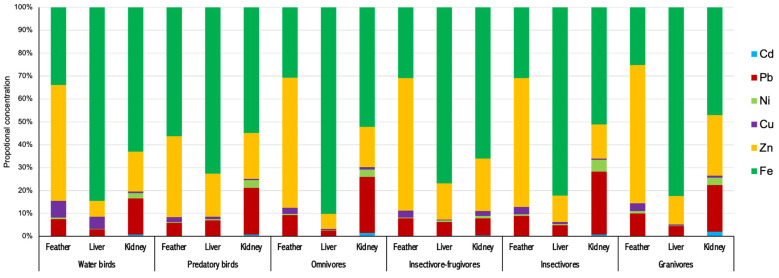
Tissue distribution of heavy metals in three bird species.

**Figure 4 animals-14-00308-f004:**
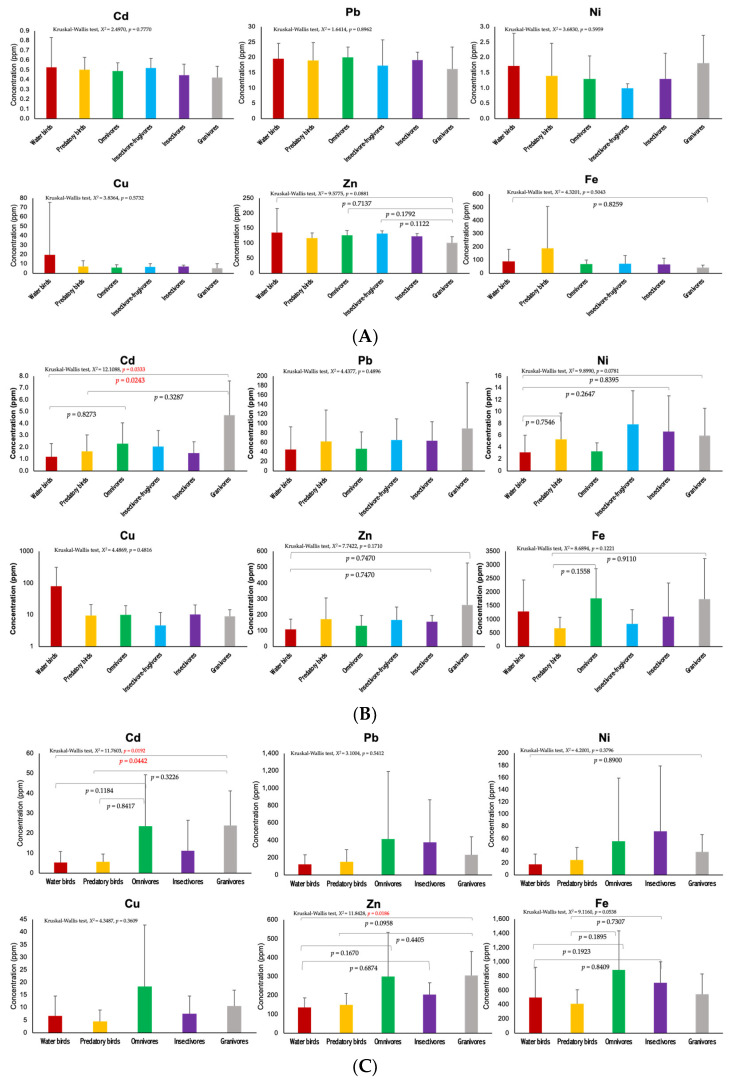
Comparison of heavy metal concentrations (mean ± SD) in feathers (**A**), livers (**B**), and kidneys (**C**) among six groups of birds. The *p* < 0.05 was considered statistically significant.

**Table 1 animals-14-00308-t001:** Sample information of birds obtained from Bang Phra Waterbird Breeding Center.

Classification	Species No.	Common Name	Scientific Name	Age	*n*	IUCN Status	Reference for Bird Classification
Water bird	1	Cinnamon Bittern	*Ixobrychus cinnamomeus*	Adult	1	LC [41]	Maneas et al. (2020) [31]
	2	Chinese Pond Heron	*Ardeola bacchus*	Adult	3	LC [41]	Boros (2021) [32]
	3	Asian Openbill	*Anastomus oscitans*	Adult	2	LC [41]	Pandiyan et al. (2022) [33]
	4	Black-crowned Night Heron	*Nycticorax nycticorax*	Adult	1	LC [41]	Boros (2021) [32]
	5	Ruddy-breasted Crake	*Porzana fusca*	Adult	1	LC [41]	Boros (2021) [32]
	6	Black-headed Ibis	*Threskiornis melanocephalus*	Adult	2	NT [41]	Boros (2021) [32]
	7	Greater Painted-snipe	*Rostratula benghalensis*	Adult	2	LC [41]	Boros (2021) [32]
	8	Striated Heron	*Butorides striata*	Adult	1	LC [41]	Boros (2021) [32]
	9	Grey-headed Swamphen	*Porphyrio porphyrio*	Adult	3	LC [41]	Yang et al. (2022) [34]
	10	Watercock	*Gallicrex cinerea*	Adult	1	LC [41]	Yang et al. (2022) [34]
	11	Red-wattled Lapwing	*Vanellus indicus*	Adult	3	LC [41]	Boros (2021) [32]
	12	Brown-headed Gull	*Larus brunnicephalus*	Adult	1	LC [41]	Boros (2021) [32]
	13	Painted Stork	*Mycteria leucocephala*	Juvenile	1	LC [41]	Boros (2021) [32]
	14	Stork-billed Kingfisher	*Pelargopsis capensis*	Adult	1	LC [41]	Maneas et al., 2020 [31]
	15	Common Kingfisher	*Alcedo atthis*	Adult	1	LC [41]	Maneas et al. (2020) [31]
Predatory bird	16	Crested Serpent Eagle	*Spilornis cheela*	Adult	1	LC [41]	Christopher et al. (2019) [35]
	17	Oriental Scops Owl	*Otus sunia*	Adult	2	LC [41]	Christopher et al. (2019) [35]
	18	Eastern Barn Owl	*Tyto alba*	Adult	1	LC [41]	Christopher et al. (2019) [35]
	19	Spotted Owlet	*Athene brama*	Adult	1	LC [41]	Christopher et al. (2019) [35]
	20	Asian Barred Owlet	*Glaucidium cuculoides*	Adult	2	LC [41]	Christopher et al. (2019) [35]
	21	Collared Scops Owl	*Otus lettia*	Adult	3	LC [41]	Christopher et al. (2019) [35]
	22	Black-winged Kite	*Elanus caeruleus*	Adult	1	LC [41]	Christopher et al. (2019) [35]
	23	Greater Coucal	*Centropus sinensis*	Adult	1	LC [41]	Shafie et al. (2023) [36]
Omnivore	24	Gurney’s Pitta	*Hydrornis gurneyi*	Adult	1	CR [41]	Round (2014) [37]
	25	Indochinese Roller	*Coracias affinis*	Adult	2	LC [41]	Ranade et al. (2023) [38]
	26	Asian Koel	*Eudynamys scolopaceus*	Adult	4	LC [41]	Yarnvudhi et al. (2022) [39]
	27	Common Myna	*Acridotheres tristis*	Adult	1	LC [41]	Yarnvudhi et al. (2022) [39]
Insectivore–frugivore	28	Lineated Barbet	*Psilopogon lineatus*	Adult	2	LC [41]	Shafie et al. (2023) [36]
	29	Coppersmith Barbet	*Megalaima haemacephala*	Adult	1	LC [41]	Shafie et al. (2023) [36]
	30	Yellow-vented Bulbul	*Pycnonotus goiavier*	Adult	1	LC [41]	Shafie et al. (2023) [36]
	31	Red-whiskered Bulbul	*Pycnonotus jocosus*	Adult	1	LC [41]	Shafie et al. (2023) [36]
Insectivore	32	Green-billed Malkoha	*Phaenicophaeus tristis*	Adult	2	LC [41]	Shafie et al. (2023) [36]
	33	Greater Racquet-tailed Drongo	*Dicrurus paradiseus*	Adult	1	LC [41]	Shafie et al. (2023) [36]
	34	Barn Swallow	*Hirundo rustica*	Juvenile	1	LC [41]	Shafie et al. (2023) [36]
	35	Large-tailed Nightjar	*Caprimulgus macrurus*	Adult	1	LC [41]	Evens et al. (2020) [40]
Granivore	36	Zebra Dove	*Geopelia striata*	Adult	3	LC [41]	Yarnvudhi et al. (2022) [39]
	37	Thick-billed Green Pigeon	*Treron curvirostra*	Adult	3	LC [41]	Yarnvudhi et al. (2022) [39]

ICUN = International Union for Conservation of Nature [41]; LC = least concern; NT = near threatened; CR = critically endangered.

**Table 2 animals-14-00308-t002:** Mean ± SD, median, and range of heavy metal concentrations (ppm) in the feather, liver, and kidney of all bird species. ND indicates elements not detected.

Metal	Feather (*n* = 60)			Liver (*n* = 50)			Kidney (*n* = 45)		
	Mean ± SD	Median	Min–Max	Mean ± SD	Median	Min–Max	Mean ± SD	Median	Min–Max
Cd	0.5 ± 0.2	0.5	0.2–1.6	1.9 ± 1.8	1.3	0.4–8.4	10.8 ± 14.9	5.4	0.6–64.1
Pb	19.0 ± 5.3	18.6	4.3–32.1	56.7 ± 54.9	32.7	14.9–254.1	215.3 ± 370.8	112.6	16.2–2303.1
Ni	1.5 ± 1.0	1.5	ND–4.5	4.5 ± 4.0	2.9	1.0–17.3	32.4 ± 55.8	16.2	1.8–311.2
Cu	11.7 ± 35.6	6.2	1.1–279.9	34.7 ± 146.0	8.8	ND–1021.8	8.9 ± 12.5	6.7	ND–71.5
Zn	125.8 ± 52.2	121.5	55.0–488.4	149.7 ± 118.0	107.1	42.3–703.7	191.8 ± 129.5	165.8	59.2–755.9
Fe	98.7 ± 157.7	55.3	16.0–1053.2	1228.9 ± 1067.3	992.3	147.1–5088.4	572.6 ± 400.9	478.4	170.7–2020.8

**Table 3 animals-14-00308-t003:** Spearman’s rank correlation coefficient (*ρ*) between feathers, livers, and kidneys for six elements. The *p* values are indicated in parentheses.

Element	Tissue	Feather	Liver	Kidney
Cd	Feather	1		
	Liver	0.1054 (*p* = 0.4665)	1	
	Kidney	0.1053 (*p* = 0.4913)	0.7988 (*p* < 0.0001)	1
Pb	Feather	1		
	Liver	0.4874 (*p* = 0.0003)	1	
	Kidney	0.2315 (*p* = 0.1259)	0.6253 (*p* < 0.0001)	1
Ni	Feather	1		
	Liver	0.0469 (*p* = 0.7464)	1	
	Kidney	0.0564 (*p* = 0.7131)	0.5468 (*p* < 0.0001)	1
Cu	Feather	1		
	Liver	0.0769 (*p* = 0.5957)	1	
	Kidney	−0.3486 (*p* = 0.0189)	0.0544 (*p* = 0.7225)	1
Zn	Feather	1		
	Liver	0.1603 (*p* = 0.2663)	1	
	Kidney	0.1034 (*p* = 0.4990)	0.5241 (*p* = 0.0002)	1
Fe	Feather	1		
	Liver	0.0182 (*p* = 0.9004)	1	
	Kidney	−0.1202 (*p* = 0.4315)	0.7988 (*p* = 0.0257)	1

## Data Availability

Data are contained within the article.

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
