# Peer review of "Species Differences and Tissue Distribution of Heavy Metal Residues in Wild Birds"

_animals, 2024, doi:10.3390/ani14020308_

Round 1

Reviewer 1 Report

Comments and Suggestions for Authors

Dear authors, your work deals with heavy metals, an environmental topic that always raises concerns. Furthermore, it involves threatened wild birds and different tissues, thereby increasing its significance. Below, I present my review:

Using keywords different from the title that also address the topic of the study can increase the visibility of your work in bibliographic searches. Therefore, it is advisable to use alternative keywords that are relevant to the study.

This comment is a mere curiosity and does not invalidate your work, but I wonder the following: Why have mercury and arsenic not been included in this study? In research on potentially toxic elements, these four elements—As, Cd, Hg, and Pb—are often studied together, and additional elements such as Fe or Ni are commonly added. Both As and Hg are highly toxic at low doses.

Line 68. Cite correctly. Pratheep et al. [] and subsequent references.

Lines 73-84. Provide a citation on this matter.

Line 93-94. I wouldn't be so categorical with this statement, as a variety of species or sources are used in environmental pollution studies. Therefore, I would adapt the statement by removing the term "more popular"; all species have their advantages and disadvantages.

Line 143-149. What post hoc test was used for multiple comparisons? If you used parametric tests, were the assumptions of normality and homoscedasticity of variances respected with their respective tests? Metal concentrations generally do not follow normal distributions, and if they don't, it might be more appropriate to use non-parametric tests (Kruskal-Wallis for ANOVA and Spearman correlation for Pearson correlation), or the data could be transformed. This is important as it could impact your statistical analysis.

Line 159. Figure 1 is visually appealing, but it should be moved higher in the text, and it would be advisable to include a more visible scale, an arrow indicating north, and coordinates.

Line 160. Table 1 provides a lot of information. However, I would adapt it to a more scientific format, and the column labeled "classification" should be placed on the left instead of the right. In general, the placement of tables and figures should align with the text.

Line 189. The concentration of Pb seems highly relevant here, as it is very elevated.

Line 210. The content of the table seems perfect, but the format should be adapted to a scientific style. Here, you indicate the use of the Tukey test as a post hoc, but it would be appropriate to mention it in the materials and methods section instead of here.

Line 211-213. This information should be in the table's footer.

Line 222. It would be advisable to increase the font size in Figure 2, as it is challenging to read, especially on the Y-axis.

Line 229. In Figure 2, it would be helpful to clarify that the letters refer to each metal and the wild bird group. The current approach is a bit confusing. The best way is as follows: "Sharing a letter indicates no differences." This way, "A" and "B" would have differences between them, while "AB" would have no differences with "A" and "B." Also, in situations where no letter is assigned, it should be indicated with the letter "A."

Line 244. This figure is excellent.

Line 296. "Biosentinels" seems like a good keyword.

Lines 331-340. Here, it would be beneficial to clearly discuss the function of each organ. For example, you talk about the liver and kidney in comparison with feathers but not about the liver compared to the kidney. For instance, when discussing iron.

Line 346. The citation needs correction.

Check the format of the references.

Comments on the Quality of English Language

No comments about it.

Author Response

Thank you for reviewing our manuscript. Please find our responses below, with revised words or sentences in the manuscript highlighted in yellow (the attached file). In addition, portions of the manuscript have been edited to improve readability.

Dear authors, your work deals with heavy metals, an environmental topic that always raises concerns. Furthermore, it involves threatened wild birds and different tissues, thereby increasing its significance. Below, I present my review:

Using keywords different from the title that also address the topic of the study can increase the visibility of your work in bibliographic searches. Therefore, it is advisable to use alternative keywords that are relevant to the study.

Keywords were changed (Line 39).

This comment is a mere curiosity and does not invalidate your work, but I wonder the following: Why have mercury and arsenic not been included in this study? In research on potentially toxic elements, these four elements—As, Cd, Hg, and Pb—are often studied together, and additional elements such as Fe or Ni are commonly added. Both As and Hg are highly toxic at low doses.

According to your suggestions, we agree that As, Cd, Hg, and Pb are important toxic metals and should be studied together. However, the studies of heavy metal contaminations in Thailand found that Cd, Pb, and Ni are more frequently reported for the highest accumulation in many environmental samples and are concerned for health effects (Aendo et al., 2022, Apilux et al., 2023, Asokbunyarat and Sirivithayapakorn, 2020, Mingkhwan and Worakhunpiset, 2018). Therefore, we would like to focus on Cd, Pb, and Ni first in this research. In addition, the research of Hg and As in birds is ongoing and will be reported with the study of toxicity further.

Aendo, P.; De Garine-Wichatitsky, M.; Mingkhwan, R.; Senachai, K.; Santativongchai, P.; Krajanglikit, P.; Tulayakul, P. Potential Health Effects of Heavy Metals and Carcinogenic Health Risk Estimation of Pb and Cd Contaminated Eggs from a Closed Gold Mine Area in Northern Thailand. Foods 2022, 11, 2791. https://doi.org/10.3390/foods11182791

Apilux, A.; Thongkam, T.; Tusai, T.; Petisiwaveth, P.; Kladsomboon, S. Determination of Heavy Metal Residues in Tropical Fruits near Industrial Estates in Rayong Province, Thailand: A Risk Assessment Study. Environment and Natural Resources Journal 2023, 21, 19-34, doi:10.32526/ennrj/21/202200146.

Asokbunyarat, V. and Sirivithayapakorn, S. Heavy Metals in Sediments and Water at the Chao Phraya River Mouth, Thailand. Thai Environmental Engineering Journal 2020, 34(3) : 33-44

Mingkhwan, R., & Worakhunpiset, S. Heavy Metal Contamination Near Industrial Estate Areas in Phra Nakhon Si Ayutthaya Province, Thailand and Human Health Risk Assessment. International journal of environmental research and public health 2018, 15(9), 1890. https://doi.org/10.3390/ijerph15091890

Line 68. Cite correctly. Pratheep et al. [] and subsequent references.

It was revised (Lines 67-68).

Lines 73-84. Provide a citation on this matter.

The references were added in Lines 77 and 78.

Line 93-94. I wouldn't be so categorical with this statement, as a variety of species or sources are used in environmental pollution studies. Therefore, I would adapt the statement by removing the term "more popular"; all species have their advantages and disadvantages.

The term "the most popular" was removed (Lines 93-94).

Line 143-149. What post hoc test was used for multiple comparisons? If you used parametric tests, were the assumptions of normality and homoscedasticity of variances respected with their respective tests? Metal concentrations generally do not follow normal distributions, and if they don't, it might be more appropriate to use non-parametric tests (Kruskal-Wallis for ANOVA and Spearman correlation for Pearson correlation), or the data could be transformed. This is important as it could impact your statistical analysis.

Thank you for pointing this out. According to your question about the post hoc test, the Tukey HSD test is used as a post hoc test for multiple comparisons. The levels of heavy metal among tissues and groups of birds were examined for normality using the Shapiro-Wilk test and determined for homogeneity of variance using Levene’s test. We found that our data was not a normal distribution, therefore the non-parametric tests (Kruskal-Wallis test and Spearman correlation) were used instead. The statistical analysis was revised (Line 186-190).

Line 159. Figure 1 is visually appealing, but it should be moved higher in the text, and it would be advisable to include a more visible scale, an arrow indicating north, and coordinates.

Figure 1 was modified (Lines 135-156).

Line 160. Table 1 provides a lot of information. However, I would adapt it to a more scientific format, and the column labeled "classification" should be placed on the left instead of the right. In general, the placement of tables and figures should align with the text.

Table 1 was revised (Line 160). All figures and tables were aligned with the text.

Line 189. The concentration of Pb seems highly relevant here, as it is very elevated.

To ensure the correctness of the information, we checked the concentration of Pb many times and found that this value is correct and observed in the kidney of common myna.

Line 210. The content of the table seems perfect, but the format should be adapted to a scientific style. Here, you indicate the use of the Tukey test as a post hoc, but it would be appropriate to mention it in the materials and methods section instead of here.

Table 2 was modified (Lines 205-208) and the use of the Tukey test as a post hoc was moved to the materials and methods section(Lines 188-189).

Line 211-213. This information should be in the table's footer.

It was moved to the table’s footer (Lines 207-208).

Line 222. It would be advisable to increase the font size in Figure 2, as it is challenging to read, especially on the Y-axis.

Figure 2 was modified and the font size was increased (Line 233-234).

Line 229. In Figure 2, it would be helpful to clarify that the letters refer to each metal and the wild bird group. The current approach is a bit confusing. The best way is as follows: "Sharing a letter indicates no differences." This way, "A" and "B" would have differences between them, while "AB" would have no differences with "A" and "B." Also, in situations where no letter is assigned, it should be indicated with the letter "A."

It was revised (Lines 233-236).

Line 244. This figure is excellent.

Line 296. "Biosentinels" seems like a good keyword.

Lines 331-340. Here, it would be beneficial to clearly discuss the function of each organ. For example, you talk about the liver and kidney in comparison with feathers but not about the liver compared to the kidney. For instance, when discussing iron.

Thank you for pointing this out. It was revised. The discussion was added in lines 345-354.

Line 346. The citation needs correction.

It was modified (Lines 359-360).

Check the format of the references.

The format of the references was checked.

Reviewer 2 Report

Comments and Suggestions for Authors

Manuscript ID: animals-2740256  

Manuscript Title: Heavy Metal Residues in Endangered Wild Birds in the Bang Phra Reservoir Nonhunting Area, Thailand: Species Differ- ences and Tissue Distribution

Journal:  Animals

The study is very meaningful and novel. The authors assess heavy metal deposits in the livers, kidneys, and feathers of endangered bird species in the Bang Phra Reservoir Nonhunting Area for the first time, as well as obtained interesting findings. For example, there was interspecies differences in metal residues in livers and kidneys but not in feathers. The topic of this MS falls within the scope of Animals. However, before it is acceptable for publication in Animals, the authors should give appropriate revise. The specific comments are at the below.

 Simple Summary

1. L 19: Revise “Pb concentrations” to “Pb concentration”.

2. L 20:Revise “the potential toxicity” to “potential toxicity.

3. L 20-21: Revise “to affect all groups of birds” to “to affect the health of birds”. Is the change right?

4. L 21:Revise “of the liver ” to “of liver.

5. L 23:Revise “The study reveals ” to “The study revealed.

6. L 24:Revise “which serves ” to “which served.

Abstract Section

7. L 27: Revise “This study aims” to “This study aimed”.

8. L 28: Revise “the tissue distribution” to “tissue distribution”.

9. L 30:Revise “in the feathers” to “in feathers”.

10. L 32:Revise “Cd levels ” to “Cd level

11.  L 32: Revise “in the kidneys” to “in kidneys”.

12. L 32: Revise “the threshold ranges” to “Pb and Cd thresholds, respectively”. Is the change right?

13. L 33: Revise “to all groups of birds” to “to the health of birds”. Is the change

14. L 33: Revise “in the livers” to “in livers”.

15. L 37-38: Revise “to control both the birds and other wildlife species” to “to reduce the threat of heavy metals to the health of both in birds and other wildlife species”. Is the change right?

Introduction Section

16. L 47: Revise “in large concentrations” to “in large concentration”.

17. L 47-49: Revise “while nonessential metals have negative effects on animals even at low concentrations such as lead (Pb), cadmium (Cd), nickel (Ni), and mercury (Hg)” to “while nonessential metals, such as lead (Pb), cadmium (Cd), nickel (Ni), and mercury (Hg), have negative effects on animals even at low concentrations.”.

18. L 55-56: Revise “using various types of samples” to “with various types of samples”.

19. L 59-61: Revise “heavy metals have been contaminated in many parts of Chonburi Province such as coastal and industrial areas, mangrove, and agricultural farms” to “many parts of Chonburi Province, such as coastal and industrial areas, mangrove, and agricultural farms, were contaminated by heavy metals”.

20. L 63: Revise “Only few studies indicate” to “Only few studies indicated”.

21. L 64-65: Revise “above the standard level were observed” to “above standard level was found”.

22. L 71: Revise “Awareness of the heavy metal pollution problem” to “The awareness of heavy metal pollution problem”.

23. L 75: Regarding “The water levels in the reservoir and environment changes” ,water levels? 

24. L 77: Revise “protect the ecosystem” to “protect ecosystem”.

25. L 78: Revise “sambar deer, civet, etc.” to “sambar deer, and civet.”.

26. L 79: Revise “is prohibited” to “are prohibited”.

27. L 88: Revise “inhalation of polluted air” to “inhaling polluted air”.

28. L 90: Revise “the lung” to “lung”.

29. L 90: Revise “skin, and feathers” to “skin, and feather”.

30. L 92-93: Revise “Cd can induce reduced” to “Cd can reduce”.

31. L 94: It is recommended to supplement Some researches also found that kidneys and livers were target organs attacked by Pb and Cd (Wang et al., 2018; Wang et al., 2016; Cui et al., 2023; Li et al., 2018)after “pollution in the environment.” to support the study.

Hao Wang et al. The Antagonistic Effect of Selenium on Lead-Induced Inflammatory Factors and Heat Shock Proteins mRNA Expression in Chicken Livers. Biological Trace Element Research, 2016, 171(2):437-444. https://doi.org/10.1007/s12011-015-0532-z. 

Xiaoyu Wang et al. Selenium protects against lead-induced apoptosis via endoplasmic reticulum stress in chicken kidneys. Biological Trace Element Research, 2018, 182(2):354-363. https://doi.org/10.1007/s12011-017-1097-9.

Jiawen Cui et al. Cadmium induced time-dependent kidney injury in common carp via mitochondrial pathway: impaired mitochondrial energy metabolism and mitochondrion-dependent apoptosis. Aquatic Toxicology, 2023, 106570. doi: 10.1016/j.aquatox.2023.106570.

 Zhuo Li et al. The contributions of miR-25-3p, oxidative stress, and heat shock protein in a complex mechanism of autophagy caused by pollutant cadmium in common carp (Cyprinus carpio L.) hepatopancreas. Environmental Pollution, 2021, 287:117554. https://doi.org/10.1016/j.envpol.2021.117554. 

32. L 96-97: Revise “using feather, liver, and kidney” to “with feathers, livers, and kidneys”.

Materials and Methods Section.

33. L 114: Revise “are shown” to “were shown”.

34. L 115: Revise “The liver, kidney, and feathers” to “The livers, kidneys, and feathers”.

35. L 117: Revise “-80oC” to “-80 °C”.

36. L 119: Revise “The sample” to “ Sample”.

37. L 122: Revise “60oC” to “60 °C”.

38. L 125: Revise “25 ml” to “25 mL”.

39. L 125: Revise “for concentrations” to “for the concentrations”.

40. L 130: Revise “the sample preparation” to “sample preparation”.

41. L 141-142: Regarding “the average ± standard error of the mean (SE).”, average or mean?

42. L 146:Revise “the feather, liver, and kidney” to “feather, liver, and kidney.

43. L 159: Revise “Location of” to “The location of”.

44. L 160: Please change table 1 to three-line table.

Results Section.

45. L 163: Revise “Differences of” to “The differences of”.

46. L 163: Revise “groups of birds” to “the groups of birds”.

47. L 164: Revise “Table 2 shows” to “Table 2 showed”.

48. L 166: Revise “in the kidney” to “in kidneys”.

49. L 167-168: Revise “there was no significant difference” to “there were no significant differences”.

50. L 169-170: Revise “the kidney (572.6 ± 59.8 ppm) and feather” to “kidneys (572.6 ± 59.8 ppm) and feathers”.

51. L 173: Revise in the feathers (Figure 2.1), livers (Figure 2.2) andto in feathers (Figure 2.1), livers (Figure 2.2), and.

52. L 175: Revise “the maximum level” to “the maximum levels”.

53. L 176: Revise “was observed” to “were found”.

54. L 177: Revise “in the feather tissue” to “in feather tissues”.

55. L 193: Revise “The relative concentrations” to “Relative concentrations”.

56. L 196-197: Please rewrite “the proportional tissue concentrations”.

57. L 205: Revise “the liver” to “liver tissues”.

58. L 207: Revise “concentrations of Pb in feathers were positively correlated with those in the liver tissues” to “concentration of Pb in feathers was positively correlated with that in livers”.

Discussion Section

59. L 310: Revise “affecting the health risk” to “posing health risk”. Is the change right?

60. L 315: Revise “has revealed” to “revealed.

 L 327: “?

Conclusion Section

61. L 375: Revise “in the liver, kidney,” to “in the livers, kidneys,.

62. L 381: Revise “the livers and feathers” to “livers and feathers”.

63. L 381: Revise “Cd levels in the kidneys” to “Cd level in kidneys”.

64. L 382: Revise “the threshold ranges” to “the thresholds of Pb and Cd”. Is the change  right?

65. L 382: Revise “the potential toxicity risks” to “potential toxicity risks”.

66. L 385-386: Regarding “some species of water birds may considerably accumulate and excrete toxic metals through feathers”, How do feathers excrete toxic heavy metals in water birds?

67. L 388: Revise “the study reveals” to “the study revealed”.

Comments on the Quality of English Language

Minor editing of English language required

Author Response

Thank you very much for your comments. Please find our responses below, with revised words or sentences in the manuscript highlighted in yellow (the attached file).  In addition, portions of the manuscript have been edited to improve readability

The study is very meaningful and novel. The authors assess heavy metal deposits in the livers, kidneys, and feathers of endangered bird species in the Bang Phra Reservoir Nonhunting Area for the first time, as well as obtained interesting findings. For example, there was interspecies differences in metal residues in livers and kidneys but not in feathers. The topic of this MS falls within the scope of Animals. However, before it is acceptable for publication in Animals, the authors should give appropriate revise. The specific comments are at the below.

 Simple Summary

  1. L 19: Revise “Pb concentrations” to “Pb concentration”.

It was revised (Line 19).

  1. L 20:Revise “the potential toxicity” to “potential toxicity”.

It was revised (Line 20).

  1. L 20-21: Revise “to affect all groups of birds” to “to affect the health of birds”. Is the change right?

Yes, it’s right. It was revised (Lines 20-21).

  1. L 21:Revise “of the liver ” to “of liver”.

It was revised (Line 21).

  1. L 23:Revise “The study reveals ” to “The study revealed”.

It was revised (Line 23).

  1. L 24:Revise “which serves ” to “which served”.

It was revised (Line 24).

Abstract Section

  1. L 27: Revise “This study aims” to “This study aimed”.

It was revised (Line 27).

  1. L 28: Revise “the tissue distribution” to “tissue distribution”.

It was revised (Line 28).

  1. L 30:Revise “in the feathers” to “in feathers”.

It was revised (Line 30).

  1. L 32:Revise “Cd levels ” to “Cd level”

It was revised (Line 32).

  1. L 32: Revise “in the kidneys” to “in kidneys”.

It was revised (Line 32).

  1. L 32: Revise “the threshold ranges” to “Pb and Cd thresholds, respectively”. Is the change right?

Yes, it’s right. It was revised (Line 32).

  1. L 33: Revise “to all groups of birds” to “to the health of birds”. Is the change

It was revised (Line 33).

  1. L 33: Revise “in the livers” to “in livers”.

It was revised (Line 33).

  1. L 37-38: Revise “to control both the birds and other wildlife species” to “to reduce the threat of heavy metals to the health of both in birds and other wildlife species”. Is the change right?

Yes, it’s right. It was revised (Lines 37-38).

Introduction Section 

  1. L 47: Revise “in large concentrations” to “in large concentration”.

It was revised (Line 47).

  1. L 47-49: Revise “while nonessential metals have negative effects on animals even at low concentrations such as lead (Pb), cadmium (Cd), nickel (Ni), and mercury (Hg)” to “while nonessential metals, such as lead (Pb), cadmium (Cd), nickel (Ni), and mercury (Hg), have negative effects on animals even at low concentrations.”.

It was revised (Lines 47-49).

  1. L 55-56: Revise “using various types of samples” to “with various types of samples”.

It was revised (Line 55-56).

  1. L 59-61: Revise “heavy metals have been contaminated in many parts of Chonburi Province such as coastal and industrial areas, mangrove, and agricultural farms” to “many parts of Chonburi Province, such as coastal and industrial areas, mangrove, and agricultural farms, were contaminated by heavy metals”.

It was revised (Lines 59-61).

  1. L 63: Revise “Only few studies indicate” to “Only few studies indicated”.

It was revised (Line 63).

  1. L 64-65: Revise “above the standard level were observed” to “above standard level was found”.

It was revised (Lines 64-65).

  1. L 71: Revise “Awareness of the heavy metal pollution problem” to “The awareness of heavy metal pollution problem”.

It was revised (Line 71).

  1. L 75: Regarding “The water levels in the reservoir and environment changes” ,“water levels”? 

It was modified to “Water levels in the reservoir changes throughout the year” (Line 75).

  1. L 77: Revise “protect the ecosystem” to “protect ecosystem”.

It was revised (Line 77).

  1. L 78: Revise “sambar deer, civet, etc.” to “sambar deer, and civet.”.

It was revised (Line 78).

  1. L 79: Revise “is prohibited” to “are prohibited”.

It was revised (Line 79).

  1. L 88: Revise “inhalation of polluted air” to “inhaling polluted air”.

It was revised (Line 88).

  1. L 90: Revise “the lung” to “lung”.

It was revised (Line 90).

  1. L 90: Revise “skin, and feathers” to “skin, and feather”.

It was revised (Line 90).

  1. L 92-93: Revise “Cd can induce reduced” to “Cd can reduce”.

It was revised (Line 92).

  1. L 94: It is recommended to supplement “Some researches also found that kidneys and livers were target organs attacked by Pb and Cd (Wang et al., 2018; Wang et al., 2016; Cui et al., 2023; Li et al., 2018)” after “pollution in the environment.” to support the study.

Hao Wang et al. The Antagonistic Effect of Selenium on Lead-Induced Inflammatory Factors and Heat Shock Proteins mRNA Expression in Chicken Livers. Biological Trace Element Research, 2016, 171(2):437-444. https://doi.org/10.1007/s12011-015-0532-z. 

Xiaoyu Wang et al. Selenium protects against lead-induced apoptosis via endoplasmic reticulum stress in chicken kidneys. Biological Trace Element Research, 2018, 182(2):354-363. https://doi.org/10.1007/s12011-017-1097-9. 

Jiawen Cui et al. Cadmium induced time-dependent kidney injury in common carp via mitochondrial pathway: impaired mitochondrial energy metabolism and mitochondrion-dependent apoptosis. Aquatic Toxicology, 2023, 106570. doi: 10.1016/j.aquatox.2023.106570.

 Zhuo Li et al. The contributions of miR-25-3p, oxidative stress, and heat shock protein in a complex mechanism of autophagy caused by pollutant cadmium in common carp (Cyprinus carpio L.) hepatopancreas. Environmental Pollution, 2021, 287:117554. https://doi.org/10.1016/j.envpol.2021.117554. 

The suggested sentence was added (Lines 94-95).

  1. L 96-97: Revise “using feather, liver, and kidney” to “with feathers, livers, and kidneys”.

It was revised (Line 97).

Materials and Methods Section.

  1. L 114: Revise “are shown” to “were shown”.

It was revised (Line 115).

  1. L 115: Revise “The liver, kidney, and feathers” to “The livers, kidneys, and feathers”.

It was revised (Line 116).

  1. L 117: Revise “-80oC” to “-80 °C”.

It was revised (Line 117).

  1. L 119: Revise “The sample” to “ Sample”.

It was revised (Line 162).

  1. L 122: Revise “60oC” to “60 °C”.

It was revised (Line 165).

  1. L 125: Revise “25 ml” to “25 mL”.

It was revised (Line 168).

  1. L 125: Revise “for concentrations” to “for the concentrations”.

It was revised (Line 168).

  1. L 130: Revise “the sample preparation” to “sample preparation”.

It was revised (Line 173).

  1. L 141-142: Regarding “the average ± standard error of the mean (SE).”, “average” or “mean”?

It was changed from “average” to “mean” (Line 184).

  1. L 146:Revise “the feather, liver, and kidney” to “feather, liver, and kidney”.

It was revised (Line 191).

  1. L 159: Revise “Location of” to “The location of”.

It was revised (Line 158).

  1. L 160: Please change table 1 to three-line table.

Table 1 was changed to the three-line table (Line 159-160).

Results Section.

  1. L 163: Revise “Differences of” to “The differences of”.

It was revised (Line 195).

  1. L 163: Revise “groups of birds” to “the groups of birds”.

It was revised (Line 195).

  1. L 164: Revise “Table 2 shows” to “Table 2 showed”.

It was revised (Line 196).

  1. L 166: Revise “in the kidney” to “in kidneys”.

It was revised (Line 198).

  1. L 167-168: Revise “there was no significant difference” to “there were no significant differences”.

It was revised (Line 199-200).

  1. L 169-170: Revise “the kidney (572.6 ± 59.8 ppm) and feather” to “kidneys (572.6 ± 59.8 ppm) and feathers”.

It was revised (Line 201-202).

  1. L 173: Revise “in the feathers (Figure 2.1), livers (Figure 2.2) and” to “in feathers (Figure 2.1), livers (Figure 2.2), and”.

It was revised (Line 211).

  1. L 175: Revise “the maximum level” to “the maximum levels”.

It was revised (Line 213).

  1. L 176: Revise “was observed” to “were found”.

It was revised (Line 214).

  1. L 177: Revise “in the feather tissue” to “in feather tissues”.

It was revised (Line 215).

  1. L 193: Revise “The relative concentrations” to “Relative concentrations”.

It was revised (Line 241).

  1. L 196-197: Please rewrite “the proportional tissue concentrations”.

It was rewrite to “relative concentrations” (Line 244-245).

  1. L 205: Revise “the liver” to “liver tissues”.

It was revised (Line 265).

  1. L 207: Revise “concentrations of Pb in feathers were positively correlated with those in the liver tissues” to “concentration of Pb in feathers was positively correlated with that in livers”.

It was revised (Line 267).

Discussion Section

  1. L 310: Revise “affecting the health risk” to “posing health risk”. Is the change right?

It was revised (Line 321).

  1. L 315: Revise “has revealed” to “revealed”.

It was revised (Line 326).

 L 327: “”?

It was revised to 2-8 ppm (Line 338).

Conclusion Section

  1. L 375: Revise “in the liver, kidney,” to “in the livers, kidneys,”.

It was revised (Line 389).

  1. L 381: Revise “the livers and feathers” to “livers and feathers”.

It was revised (Line 395).

  1. L 381: Revise “Cd levels in the kidneys” to “Cd level in kidneys”.

It was revised (Line 395).

  1. L 382: Revise “the threshold ranges” to “the thresholds of Pb and Cd”. Is the change  right?

Yes, it’s right. It was revised (Line 396).

  1. L 382: Revise “the potential toxicity risks” to “potential toxicity risks”.

It was revised (Line 396).

  1. L 385-386: Regarding “some species of water birds may considerably accumulate and excrete toxic metals through feathers”, How do feathers excrete toxic heavy metals in water birds?

According to references (Jaspers et al., 2004, Pandiyan et al., 2020), previous studies found that birds excrete heavy metals into growing feathers by molting.

Jaspers, V., Dauwe, T., Pinxten, R., Bervoets, L., Blust, R., & Eens, M. (2004). The importance of exogenous contamination on heavy metal levels in bird feathers. A field experiment with free-living great tits, Parus major. Journal of environmental monitoring : JEM, 6(4), 356–360. https://doi.org/10.1039/b314919f

Pandiyan, J., Jagadheesan, R., Karthikeyan, G. et al. Probing of heavy metals in the feathers of shorebirds of Central Asian Flyway wintering grounds. Sci Rep 10, 22118 (2020). https://doi.org/10.1038/s41598-020-79029-z

  1. L 388: Revise “the study reveals” to “the study revealed”.

It was revised (Line 402).

Reviewer 3 Report

Comments and Suggestions for Authors

The paper sent for review is a very interesting study of the presence of selected metals in the tissues of birds living in the habitat defined as a reserve. The authors performed a study of a wide range of species differing in physiological characteristics and food base. Therefore, the study carried out is valuable because it provides comprehensive information, necessary from the point of view of assessing the quality of the environment studied and its impact on birds inhabiting it. The article is of very good quality and basically has no weaknesses that require excessive correction. The analytical procedures and statistical methods used are not objectionable. I understand that the statistical methods used are due to the parametric distribution of the data, which was not signaled in the text of the paper and could have resounded in the print version. Nevertheless, this is not a prerequisite for publishing a paper. The material and methods section describes the birds from which tissues were collected. They were examined by veterinary pathologists and, as I understood from the text, the deaths of all of them were due to accidents. Nevertheless, the question should be asked whether they had other pathologies, namely, whether the organs sampled were macroscopically altered in a way that proved the presence of chronic poisoning or infectious diseases? This question seems particularly pertinent to me in view of the high levels of Pb and Cd in the kidneys of the specimens studied, which are well above the reference values. Nevertheless, I believe that the work should be published because it contains valuable data.

Author Response

Thank you for reviewing our manuscript. Please find our responses below, with revised words or sentences in the manuscript highlighted in yellow (the attached file).  In addition, portions of the manuscript have been edited to improve readability.

According to your questions about whether they had other pathologies, namely, whether the organs sampled were macroscopically altered in a way that proved the presence of chronic poisoning or infectious diseases? The observation of pathological changes is important for the study of heavy metal toxicity, but this is out of our scope for this research. Most of the birds were not observed with any gross lesions or histopathological lesions by veterinarians because these birds do not have the owners and they do not have any special requests for necropsy. After veterinarians got the carcasses, they collected livers, kidneys, and feathers as we requested and then kept those bird carcasses in the freezer immediately. However, the authors totally agree with your suggestions. The evaluation of macroscopic and microscopic lesions in these birds is an interesting point for toxicity assessment that we want to study further.

Round 2

Reviewer 1 Report

Comments and Suggestions for Authors

Dear authors, I believe the majority of the comments I provided in my review have been addressed. However, you have noted the non-normality in your data, so statistical tests must be applied appropriately. I am concerned that Figure 4 appears identical when using a different statistical test.

Lines 67-68. When citing in this manner, the publication year is excluded, i.e., it should be Meewatana [16], whereas Ittiporn et al. [15]. The same applies to other citations making this error.

Line 185. SEM is correct, not SE. This stands for standard error of the mean. Perhaps it would be more appropriate to use standard deviation (SD).

Lines 186-188. The Tukey test and the Kruskal-Wallis test are statistical methods designed for different situations and types of data, so they are not used together in the same test. Perhaps the Mann-Whitney test would be more suitable. Remember that when conducting multiple comparisons, there is a risk of increasing Type I error, so it is important to adjust p-values using correction methods, such as the Bonferroni method.

My comment about the Pb concentration was not about its correctness. I simply mentioned that it seems to be a high level and is relevant to your research.

Line 217. Correct the mention of the Tukey test as a post hoc.

Line 289. Figure 4. I have carefully examined Figure 4, and it is exactly the same as in the previous version. Applying different statistical tests makes it highly unlikely that the result would be exactly the same. Please clarify this.

Author Response

Thank you very much for your review with comments. Please find our responses below, with revised words or sentences in the manuscript highlighted in yellow

Reviewer 1

Dear authors, I believe the majority of the comments I provided in my review have been addressed. However, you have noted the non-normality in your data, so statistical tests must be applied appropriately. I am concerned that Figure 4 appears identical when using a different statistical test.

Thank you very much for your concern, Figure 4 and the results of correlations were revised using a Spearman’s correlation.

Lines 67-68. When citing in this manner, the publication year is excluded, i.e., it should be Meewatana [16], whereas Ittiporn et al. [15]. The same applies to other citations making this error.

The references were modified.

Line 185. SEM is correct, not SE. This stands for standard error of the mean. Perhaps it would be more appropriate to use standard deviation (SD).

It was corrected to SEM instead of SE.

Lines 186-188. The Tukey test and the Kruskal-Wallis test are statistical methods designed for different situations and types of data, so they are not used together in the same test. Perhaps the Mann-Whitney test would be more suitable. Remember that when conducting multiple comparisons, there is a risk of increasing Type I error, so it is important to adjust p-values using correction methods, such as the Bonferroni method.

Thank you very much for your valuable suggestions, the data was analyzed using Kruskal-Wallis test and followed by Mann-Whitney test (post hoc test).

My comment about the Pb concentration was not about its correctness. I simply mentioned that it seems to be a high level and is relevant to your research.

Thank you for your comment.

Line 217. Correct the mention of the Tukey test as a post hoc.

It was revised.

Line 289. Figure 4. I have carefully examined Figure 4, and it is exactly the same as in the previous version. Applying different statistical tests makes it highly unlikely that the result would be exactly the same. Please clarify this.

Figure 4 and some results related to the correlation (line 263-270) were modified using a Spearman’s correlation.

Reviewer 2 Report

Comments and Suggestions for Authors

Manuscript ID: animals-2740256-revision  

Manuscript Title: Heavy Metal Residues in Endangered Wild Birds in the Bang Phra Reservoir Nonhunting Area, Thailand: Species Differ- ences and Tissue Distribution

Journal: Animals

The authors revised carefully the manuscript according to the requirements of the reviewer, and the quality of the revised manuscript has been improved, but there are still some minor errors that need to be corrected. The specific comments are as follow.

 Simple Summary

1. L 24: Revise “the metal pollution” to “metal pollution”.

Abstract Section

2. L 31: Revise “Pb concentrations” to “Pb concentration”.

3. L 31: Revise “in the livers” to “in livers”.

4. L 32:Revise “the potential toxic” to “potential toxic.

Introduction Section

5. L 64: Revise “ Hg concentrations” to “Hg concentration”...

6. L 85: Revise “in the ecosystem” to “in ecosystem”.

7. L 90: Revise “the lung” to “lung”.

8. L 92: Revise “the nervous system” to “nervous system”.

Materials and Methods Section.

9. L 184: Revise “are presented” to “were presented”.

Results Section.

10. L 207: Revise “related to” to “were related to”.

11. L 208: Revise “range of the” to “the range of”.

12. L 211: Revise “in the liver” to “in liver”.

13. L 212: Revise “in the liver” to “in livers”.

14. L 213: Revise “in the liver” to “in livers”.

15. L 215: Revise “in the Cu” to “in Cu”.

16. L 222: Revise “Figure 2 presents” to “Figure 2 presented”.

17. L 228: Revise “In the liver” to “In livers”.

18. L 233: Revise “the highest accumulation” to “the highest accumulations”.

19. L 234: Revise “was observed” to “were observed”.

20. L 235: Revise “In the kidney tissue” to “In kidney tissues”.

21. L 240: Revise “the highest accumulation” to “the highest accumulations”.

22. L 241: Revise “was found” to “were found”.

23. L 254: Revise “are shown” to “was shown”.

24. L 254: Revise “in the Figure 3” to “in Figure 3”.

25. L 258: Revise “in the liver” to “in livers”.

26. L 259: Revise “in the feathers” to “in feathers”.

27. L 261: Revise “in the kidney tissue of” to “in kidney tissues of”.

28. L 274: Revise “Relationship of” to “The relationship of”.

29. L 275: Revise “Figure 4 presents the” to “Figure 4 presented”.

30. L 277: Revise “the accumulation of” to “the accumulations of”.

 Discussion Section

31. L 333: Revise “the pollutants” to “pollutants.

32. L 337: Revise “Although birds” to “Although the birds”.

33. L 356: Revise “are contaminated” to “were contaminated.

34. L 358: Revise “in the livers” to “in livers.

35. L 361: Revise “in the water” to “in water.

36. L 365: Revise “in the livers” to “in livers.

37. L 367: Revise “in the kidney” to “in kidneys.

38. L 382: Revise “study suggests” to “study suggested”.

39. L 382: Revise “the liver or kidney” to “liver or kidney.

40. L 386: Revise “the Pb accumulation” to “Pb accumulation”.

41. L 387: Revise “the significant correlation” to “significant correlation”.

42. L 389: Revise “suggest that” to “suggested that.

43. L 390: Revise “the detrimental consequences” to “detrimental consequences”.

44. L 404-405: Revise “in the feathers” to “in feathers.

45. L 409: Revise “in the feathers” to “in feathers.

46. L 409: Revise “are similar to” to “were similar to.

Comments on the Quality of English Language

Minor editing of English language required

Author Response

Thank you very much for your letter with comments. Please find our responses below, with revised words or sentences in the manuscript highlighted in yellow

Reviewer 2

The authors revised carefully the manuscript according to the requirements of the reviewer, and the quality of the revised manuscript has been improved, but there are still some minor errors that need to be corrected. The specific comments are as follow. 

Thank you very much for your comments.

Simple Summary

  1. L 24: Revise “the metal pollution” to “metal pollution”.

It was revised.

Abstract Section

  1. L 31: Revise “Pb concentrations” to “Pb concentration”.

It was modified.

  1. L 31: Revise “in the livers” to “in livers”.

It was revised.

  1. L 32:Revise “the potential toxic” to “potential toxic”.

It was revised.

Introduction Section 

  1. L 64: Revise “ Hg concentrations” to “Hg concentration”...

It was revised.

  1. L 85: Revise “in the ecosystem” to “in ecosystem”.

It was revised.

  1. L 90: Revise “the lung” to “lung”.

It was revised.

  1. L 92: Revise “the nervous system” to “nervous system”.

It was modified.

Materials and Methods Section.

  1. L 184: Revise “are presented” to “were presented”.

It was revised.

Results Section.

  1. L 207: Revise “related to” to “were related to”.

It was modified (line 195).

  1. L 208: Revise “range of the” to “the range of”.

It was revised (line 196).

  1. L 211: Revise “in the liver” to “in liver”.

It was revised (line 199).

  1. L 212: Revise “in the liver” to “in livers”.

It was revised (line 200).

  1. L 213: Revise “in the liver” to “in livers”.

It was revised (201).

  1. L 215: Revise “in the Cu” to “in Cu”.

It was revised (line 202).

  1. L 222: Revise “Figure 2 presents” to “Figure 2 presented”.

It was revised (line 210).

  1. L 228: Revise “In the liver” to “In livers”.

It was revised (line 216).

  1. L 233: Revise “the highest accumulation” to “the highest accumulations”.

It was revised (line 221).

  1. L 234: Revise “was observed” to “were observed”.

It was revised (line 222).

  1. L 235: Revise “In the kidney tissue” to “In kidney tissues”.

It was revised (line 223).

  1. L 240: Revise “the highest accumulation” to “the highest accumulations”.

It was revised (line 228).

  1. L 241: Revise “was found” to “were found”.

It was revised (line 228).

  1. L 254: Revise “are shown” to “was shown”.

It was revised (line 242).

  1. L 254: Revise “in the Figure 3” to “in Figure 3”.

It was revised (line 242).

  1. L 258: Revise “in the liver” to “in livers”.

It was revised (line 246).

  1. L 259: Revise “in the feathers” to “in feathers”.

It was modified (line 247).

  1. L 261: Revise “in the kidney tissue of” to “in kidney tissues of”.

It was revised (line 249).

  1. L 274: Revise “Relationship of” to “The relationship of”.

It was revised (line 262).

  1. L 275: Revise “Figure 4 presents the” to “Figure 4 presented”.

It was revised (line 263).

  1. L 277: Revise “the accumulation of” to “the accumulations of”.

It was revised (line 265).

 Discussion Section

  1. L 333: Revise “the pollutants” to “pollutants”.

It was revised (line 304).

  1. L 337: Revise “Although birds” to “Although the birds”.

It was revised (line 308).

  1. L 356: Revise “are contaminated” to “were contaminated”.

It was revised (line 327).

  1. L 358: Revise “in the livers” to “in livers”.

It was revised (line 329).

  1. L 361: Revise “in the water” to “in water”.

It was revised (line 332).

  1. L 365: Revise “in the livers” to “in livers”.

It was revised (line 336).

  1. L 367: Revise “in the kidney” to “in kidneys”.

It was revised (line 338).

  1. L 382: Revise “study suggests” to “study suggested”.

It was revised (line 352).

  1. L 382: Revise “the liver or kidney” to “liver or kidney”.

It was revised (line 352).

  1. L 386: Revise “the Pb accumulation” to “Pb accumulation”.

It was revised (line 356).

  1. L 387: Revise “the significant correlation” to “significant correlation”.

It was revised (line 357).

  1. L 389: Revise “suggest that” to “suggested that”.

It was revised (line 359).

  1. L 390: Revise “the detrimental consequences” to “detrimental consequences”.

It was revised (line 360).

  1. L 404-405: Revise “in the feathers” to “in feathers”.

It was revised (line 374).

  1. L 409: Revise “in the feathers” to “in feathers”.

It was revised (line 378).

  1. L 409: Revise “are similar to” to “were similar to”.

It was revised (line 379).

Round 3

Reviewer 1 Report

Comments and Suggestions for Authors

Except for the fact that It would be more convenient to use Standar Deviation (SD) in the tables instead of the SEM, as I mentioned in my previous review, I believe that the manuscrit has been sufficiently improved for publication.

Comments on the Quality of English Language

I have no comments

Author Response

Thank you very much for your suggestion

Except for the fact that It would be more convenient to use Standar Deviation (SD) in the tables instead of the SEM, as I mentioned in my previous review, I believe that the manuscrit has been sufficiently improved for publication.

According to your comment, the Table 2, Figure 2, and some text in the yellow highlight (line 184, 196-202) in the attached manuscript were revised from SEM values to SD values.
